# SymDQN: Symbolic Knowledge and Reasoning in Neural Network-based Reinforcement Learning

**Ivo Amador**                                                        IVOAMADOR@GMAIL.COM

**Nina Gierasimczuk**                                                      NIGI@DTU.DK

*Department of Applied Mathematics and Computer Science, Technical University of Denmark*

Editors: Leilani H. Gilpin, Eleonora Giunchiglia, Pascal Hitzler, and Emile van Krieken

## Abstract

We propose a learning architecture that allows symbolic control and guidance in reinforcement learning with deep neural networks. We introduce `SymDQN`, a novel modular approach that augments the existing Dueling Deep Q-Networks (`DuelDQN`) architecture with modules based on the neuro-symbolic framework of Logic Tensor Networks (LTNs). The modules guide action policy learning and allow reinforcement learning agents to display behavior consistent with reasoning about the environment. Our experiment is an ablation study performed on the modules. It is conducted in a reinforcement learning environment of a 5x5 grid navigated by an agent that encounters various shapes, each associated with a given reward. The underlying `DuelDQN` attempts to learn the optimal behavior of the agent in this environment, while the modules facilitate shape recognition and reward prediction. We show that our architecture significantly improves learning, both in terms of performance and the precision of the agent. The modularity of `SymDQN` allows reflecting on the intricacies and complexities of combining neural and symbolic approaches in reinforcement learning.

## 1. Introduction

Despite its rapidly growing impact on society, Artificial Intelligence technologies are tormented by reliability issues, such as lack of interpretability, propagation of biases, difficulty in generalizing across domains, and susceptibility to adversarial attacks. A possible way towards more interpretable, controlled and guided algorithms leads through the field of neuro-symbolic AI, which explores new ways of integrating symbolic, logic-based knowledge in neural networks (NNs). In particular, the framework of Logic Tensor Networks (LTNs, for short) (Serafini and d'Avila Garcez, 2016; Badreddine et al., 2022) enhances learning by interpreting first-order logic formulas concretely on data used by NNs algorithms. Such formulas express properties of data and, given a fuzzy semantics, can be integrated into the loss function, thus guiding the learning process.

In this paper, we apply LTNs to a reinforcement learning problem. By integrating LTNs in the training process, our learning agent uses logic to learn the structure of the environment, to predict how different objects in the environment interact with each other, and to guide its actions by performing elementary reasoning about rewards. We investigate how such integration affects learning performance of a robust, established and well-studied framework of Dueling Deep Q-Network (`DuelDQN`, for short) (Wang et al., 2016). The structure of the paper is as follows. In Section 2 we briefly recall Logic Tensor Networks and elements of the underlying Real Logic. In Section 3 we introduce our methodology: the `SymDQN` architecture and its training process. We follow up with the presentation of the experiment in Section 4. We discuss of the results in Section 5. Section 6 concludes and outlines directions for future work.

**Related Work** Since its conception, the framework of LTNs has been applied in various domains. In computer vision, LTNs were used to inject prior knowledge about object relationships and properties, improving interpretability and accuracy in object detection (Donadello et al., 2017). Their addition to convolutional neural networks improves the robustness on noisy data (Manigrasso et al., 2021). They enhance the accuracy in reasoning tasks in open-world and closed-world scenarios (Wagner and d'Avila Garcez, 2022). In (Bianchi and Hitzler, 2019), LTNs are leveraged for deductive reasoning tasks. Finally, in learning by reinforcement LTNs were used to integrate prior knowledge into reinforcement learning agents improving both the learning rate and robustness to environmental changes (Badreddine and Spranger, 2019). The latter work is similar to ours in the choice of tools, but it differs in its methodology. In (Badreddine and Spranger, 2019), LTN is a separate pre-training module which interacts with `DuelDQN` only by creating inputs. In contrast, our `SymDQN` integrates LTN in the training process (making it learn alongside `DuelDQN`). Our work uses logic to adjust elements of a reinforcement learning framework. In that, it is related to reward shaping approaches, where the learner is given external symbolic information about the environment, e.g., in the form of linear time logic formulas (also known as restraining bolts) in (Giacomo et al., 2019) or of an induced automaton in (Furelos-Blanco et al., 2021). In a way, the LTN approach is similar: logical formulas adjust the reinforcement learning process. However, our technique is a more distributed form of reward shaping. First, the formulas of Real Logic are used as guides to obtain a symbolic representation of the environment, then to predict immediate rewards from encountering the objects of the environment. Finally, a logical formula is used to help the learner align the $q$-values (the agent's long term policy) with the predicted immediate rewards of symbolically represented objects. In other words, we restrain the reinforcement learner by expecting it to reason about its behavior as it learns, and we investigate the impact of this restriction on learning precision and performance.

## 2. Real Logic

Real Logic is the basis of the functioning of LTNs. In this section we provide a rudimentary introduction (for a full exposition consult (Badreddine et al., 2022)). Let us start with a simple example.

**Example 1** *Consider two datasets: a data set of humans (with two features: age and gender), and a dataset of pets (with three features: height, weight and color). Assume that Alice appears in the data set of humans (for instance as a five year old female), and Max and Mittens are listed in a dataset of pets. To be able to talk about Alice, Max and Mittens, we need a logical language that includes constants referring to objects (particular rows of the datasets). Note that such constants can be of different types—in our example humans consists of two, while pets are composed of three features.*

The signature of the language of Real Logic $\mathcal{L}$ contains a set $\mathcal{C}$ of constant symbols, a set $\mathcal{F}$ of function symbols, a set $\mathcal{P}$ of predicate symbols, and a set $\mathcal{X}$ of variable symbols. Let $\mathcal{D}$ be a non-empty set of domain symbols (that represent types). Domain symbols are used by functions $\mathbf{D}$, $\mathbf{D_{in}}$, and $\mathbf{D_{out}}$ which for a given element of the signature output its type, in the following way. $\mathbf{D} : \mathcal{X} \cup \mathcal{C} \to \mathcal{D}$ specifies the types for variables and constants;

$\mathbf{D_{in}} : \mathcal{F} \cup \mathcal{P} \to \mathcal{D}^*$ specifies the types of the sequence of arguments allowed by function and predicate symbols ( $\mathcal{D}^*$ stands for the set of all finite sequences of elements from $\mathcal{D}$); $\mathbf{D_{out}} : \mathcal{F} \to \mathcal{D}$ specifies the type of the range of a function symbol.

**Example 2** *Continuing Example 1, let the language of pet-ownership $\mathcal{L}_{pets}$ have the signature consisting of the set of constants $\mathcal{C} = \{ALICE, MAX, MIT\}$, a set of function symbols $\mathcal{F} = \{OWNER\}$, a set of predicate symbols $\mathcal{P} = \{ISOWNER\}$, and two variable symbols $\mathcal{X} = \{PET, PERSON\}$. Further, we have two domain symbols, one for the domain of humans and one for pets, $\mathcal{D} = \{H, P\}$. Then, our domain functions can be defined in the following way. $\mathbf{D}(ALICE) = H$ (Alice is a constant of type $H$), $\mathbf{D}(MAX) = \mathbf{D}(MIT) = P$ (Max and Mittens are of type $P$). Further, each dataset will have its own variable: $\mathbf{D}(PET) = P$, $\mathbf{D}(PERSON) = H$. We also need to specify inputs for predicates: $\mathbf{D_{in}}(ISOWNER) = HP$ (ISOWNER is a predicate taking two arguments, a human and a pet). Finally, for functions, we need both the input and the output types: $\mathbf{D_{in}}(OWNER) = P$, and $\mathbf{D_{out}}(OWNER) = H$ (OWNER takes as input a pet and outputs the human who owns it).*

The language of Real Logic corresponds to first-order logic, and so it allows for more complex expressions. The set of terms consists of constants, variables, and function symbols and is defined in the following way: each $t \in X \cup C$ is a term of domain $\mathbf{D}(t)$; if $t_1, \ldots, t_n$ are terms, then $t_1 \ldots t_n$ is a term of the domain $\mathbf{D}(t_1)...\mathbf{D}(t_n)$; if $t$ is a term of the domain $\mathbf{D_{in}}(f)$, then $f(t)$ is a term of the domain $\mathbf{D_{out}}(f)$. Finally, the formulae of Real Logic are as follows: $t_1 = t_2$ is an atomic formula for any terms $t_1$ and $t_2$ with $\mathbf{D}(t_1) = \mathbf{D}(t_2)$; $P(t)$ is an atomic formula if $\mathbf{D}(t) = \mathbf{D_{in}}(P)$; if $\varphi$ and $\psi$ are formulae and $x_1, \ldots, x_n$ are variable symbols, then $\neg\varphi$, $\varphi \wedge \psi$, $\varphi \vee \psi$, $\varphi \to \psi$, $\varphi \leftrightarrow \psi$, $\forall x_1 \ldots x_n \varphi$ and $\exists x_1 \ldots x_n \varphi$ are formulae. Let us now turn to the semantics of Real Logic. Domain symbols allow grounding the logic in numerical, data-driven representations—to be precise, Real Logic is grounded in tensors in the field of real numbers. Tensor grounding is the key concept that allows the interplay of Real Logic with Neural Networks. It refers to the process of mapping high-level symbols to tensor representations, allowing integration of reasoning and differentiable functions. A grounding $\mathcal{G}$ assigns to each domain symbol $D \in \mathcal{D}$ a set of tensors $\mathcal{G}(D) \subseteq \bigcup_{n_1 \ldots n_d \in \mathbb{N}^*} \mathbb{R}^{n_1 \times \ldots \times n_d}$. For every $D_1 \ldots D_n \in \mathcal{D}^*$, $\mathcal{G}(D_1 \ldots D_n) = \mathcal{G}(D_1) \times \ldots \times \mathcal{G}(D_n)$. Given a language $\mathcal{L}$, a grounding $\mathcal{G}$ of $\mathcal{L}$ assigns to each constant symbol $c$, a tensor $\mathcal{G}(c)$ in the domain $\mathcal{G}(\mathbf{D}(c))$; to a variable $x$ it assigns a finite sequence of tensors $d_1 \ldots d_k$, each in $\mathcal{G}(\mathbf{D}(x))$, representing the instances of $x$; to a function symbol $f$ it assigns a function taking tensors from $\mathcal{G}(\mathbf{D_{in}}(f))$ as input, and producing a tensor in $\mathcal{G}(\mathbf{D_{out}}(f))$ as output; to a predicate symbol $P$ it assigns a function taking tensors from $\mathcal{G}(\mathbf{D_{in}}(P))$ as input, and producing a truth-degree in the interval $[0, 1]$ as output. In other words, $\mathcal{G}$ assigns to a variable a concatenation of instances in the domain of the variable. The treatment of free variables in Real Logic is analogous, departing from the usual interpretation of free variables in FOL. Thus, the application of functions and predicates to terms with free variables results in point-wise application of the function or predicate to the string representing all instances of the variable (see p. 5 of (Badreddine et al., 2022) for examples). Semantically, logical connectives are fuzzy operators applied recursively to the suitable subformulas: conjunction is a t-norm, disjunction is a t-conorm, and for implication and negation—its fuzzy correspondents. The semantics for formulae with quantifiers ($\forall$ and $\exists$) is given by symmetric and continuous aggregation operators

$Agg : \bigcup_{n \in N}[0,1]^n \to [0,1]$. Intuitively, quantifiers reduce the dimensions associated with the quantified variables.

**Example 3** *Continuing our running example, we could enrich our signature with predicates Dog and Cat. Then, $Dog(Max)$ might return $0.8$, while $Dog(Mit)$ might return $0.3$, indicating that Max is likely a dog, while Mittens is not. In practice, the truth degrees for these atomic formulas could be obtained for example by a Multi-layer Perceptron (MLP), followed by a sigmoid function, taking the object's features as input and returning a value in $[0,1]$. For a new function symbol age, $age(Max)$ could be an MLP, taking Max's features, and outputting a scalar representing Max's age. An example of a formula could be $Dog(Max) \lor Cat(Max)$, which could return $0.95$ indicating that Max is almost certainly either a dog or a cat. A formula with a universal quantifier could be used to express that Alice owns all of the cats $\forall pet(Cat(pet) \to isOwner(Alice, pet))$.*

Real Logic allows some flexibility in the choice of appropriate fuzzy operators for the semantics of connectives and quantifiers. However, note that not all fuzzy operators are suitable for differentiable learning (van Krieken et al., 2022). In Appendix B of (Badreddine et al., 2022), the authors discuss some particularly suitable fuzzy operators. In this work, we follow their recommendation and adopt the Product Real Logic semantics (product t-norm for conjunction, standard negation, the Reichenbach implication, p-mean for the existential, and p-mean error for the universal quantifier). LTNs make use of Real Logic—they learns parameters that maximize the aggregated satisfiability of the formulas in the so-called knowledge base containing formulas of Real Logic. The framework is the basis of the PyTorch implementation of the LTN framework, known as LTNtorch library (Carraro, 2022). In our experiments we make substantial use of that tool.

## 3. Methodology

The environment used for the experiments was a custom Gymnasium (Towers et al., 2024) environment `ShapesGridEnv` designed for the experiments in (Badreddine and Spranger, 2019), see Fig. 1. The game is played on an image showing a 5x5 grid with cells occupied by one agent, represented by the symbol '+', and a number of other objects: circles, squares, and crosses. The agent moves across the board (up, right, down, left) and when it enters a space occupied by an object, it 'consumes' that object. Each object shape is associated with a reward. The agent's goal is to maximize its cumulative reward throughout an episode. An episode terminates when either all shapes with positive reward have been consumed, or when a predefined number of steps has been reached.

We chose this environment because of its simplicity, and because it allows comparing our setting with that of (Badreddine and Spranger, 2019). The environment is very flexible in its parameters: density (the minimum and maximum amount of shapes initiated, in our case max is 18), rewards (in our case the reward for a cross is $+1$, for a circle is -1 and for a square is 0), colors (in our case the background is white and objects are black), episode maximum length (for us it is 50). Altering the environment configurations allows investigating the adaptability of the learner in (Badreddine and Spranger, 2019).

A suitable approach to learning to play such a game could be the existing Dueling Deep Q-Network (`DuelDQN`) (Wang et al., 2015). The architecture is composed of several

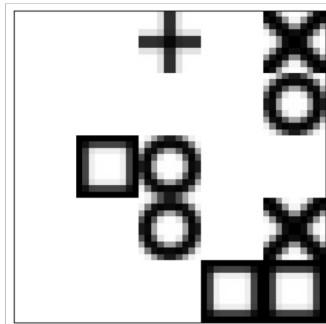

Figure 1: `ShapesGridEnv` environment

convolutional layers, which extract relevant features from the raw image input, and then pass them to the two main components, a Value Stream and an Advantage Stream (see Fig. 3 in the Appendix). The Value Stream estimates how good it is for the agent to be in the given state, while the Advantage Stream estimates how good it is to perform each action in that given state. The two streams are then combined to calculate the final output. Our starting point is `DuelDQN` architecture which we extend with new symbolic, cognitively-motivated components: shape recognition (`ShapeRecognizer`), reward prediction (`RewardPredictor`), action reasoning (`ActionReasoner`), and action filtering (`ActionFilter`).

### 3.1. Shape Recognizer

The function of `ShapeRecognizer` is to estimate the likelihood of a certain observation to be of a given unique kind. `ShapeRecognizer` is comprised of one pre-processing function, which divides the initial raw image into 25 patches. Each patch is then processed by a Convolutional Neural Network (CNN), which then outputs a 5-dimensional tensor.

The numbers chosen for the number of patches and the output dimension are an instance of soft knowledge injection, as the environment represents a 5x5 grid, and dividing it into 25 patches immediately separates the content of each cell in the grid. As for the output size, 5 is the number of different objects that each patch might contain: empty, agent, circle, cross, square. This allows the agent to perform a multi-label classification on each object type, and give it a symbolic understanding of the different entities in the environment (by labeling them by types). Given the simple nature of the `ShapesGridEnv`, representing the environment is very easy. The state is composed of 25 positions, with each position being occupied by one of five shapes (empty, agent, circle, square, cross), which results in the state space of size $25^5$. To generate this representation, we start by instantiating five one-hot representations of the classes, which are stored in the LTN Variable *shape_types*. Then, for each state that the agent is in, it keeps in memory all the different patches that it has seen and a list of all the patches that are present in the current state.

Once the variables have been set up, the `ShapeRecognizer` module can be used to estimate the likelihood of a grid cell containing a given unique shape. To guide its learning, the aggregated satisfiability of three axioms is maximized. The axioms represent the first

instance of *actual* knowledge injection in the system:

$$\forall s \; \exists l \; IS(s, l) \tag{A1}$$

$$\neg \exists s \; l_1 \; l_2 \; (IS(s, l_1) \land IS(s, l_2) \land (l_1 \neq l_2)) \tag{A2}$$

$$\forall s_1 \; s_2 \; l \; ((IS(s_1, l) \land (s_1 \neq s_2)) \to \neg IS(s_2, l) \tag{A3}$$

In the above formulas, $s$ stands for a shape, $l$ stands for a label and $IS(x, y)$ stands for $x$ has label $y$. A1 says that every shape has a label; A2 says that no shape has two different labels; A3 says that different shapes cannot have the same label. At each step of every episode, the aggregated satisfaction (truth) degree of these axioms is calculated, and its inverse, $1 - AggSat(A1, A2, A3)$, is used as a loss to train the agent.

Intuitively, `ShapeRecognizer` gives the learner a way to distinguish between different shapes. In that, our approach is somewhat similar to the framework of semi-supervised symbol grounding (Umili et al., 2023).

### 3.2. Reward Predictor

Once the environment is symbolically represented, we will make the agent understand the properties of different objects and their dynamics. The only truly dynamic element in the environment is the agent itself—nothing else moves. The agent can move to a cell that was previously occupied by a different shape, which results in the shape being consumed, and the agent being rewarded with the value of the respective shape. Hence, there are three key pieces of knowledge that the learner must acquire to successfully navigate the environment. It must identify which shape represents the agent, it must understand how each action influences the future position of the agent, and it must associate each shape with its respective reward. The task of self-recognition can be approached in numerous ways, depending on the information we have about the environment, and on our understanding of its dynamics. Leveraging the `ShapeRecognizer`, in each episode we count the occurrences of each shape in the environment. The agent is identified as the only shape that has a constant, and equal to one, number of appearances in the environment. This step demonstrates a specific advantage of using the neuro-symbolic framework. Our reinforcement learning agent is now equipped with memory of the previous states of the environment (i.e., the count of shapes) which can then be used to make decisions or to further process symbolic knowledge. Understanding the impact of different actions is crucial for the agent to make informed decisions in the environment. Each action represents a direction (up, right, down, left) and taking the action will lead to one of two outcomes: if the agent is at the edge of the environment and attempts to move against the edge, it will remain in the same cell, otherwise, the agent will move one cell in the given direction. Given the simplicity of this dynamics, a function has been defined that takes as input a position and an action, outputting the resulting position.

Our `RewardPredictor` is a Multi-layer Perceptron (MLP), which takes as input the prediction of `ShapeRecognizer` and outputs one scalar. The module trains to predict the reward associated with each shape, using the symbolic representations generated by the `ShapeRecognizer` paired with high-level reasoning on the training procedure. This module intends to give the agent a way of knowing on a high level the reward associated with any

shape, and consequently with any action. In reinforcement learning environments, agents learn action policies by maximizing their expected rewards. When building an agent that symbolically represents and reasons about the environment, one of the key elements is the agent's ability to understand how to obtain rewards. Given that the agent has the capability of identifying the shapes in the grid, recognizing its own shape, and calculating the position it will take given an action, it can now determine the shape that will be 'consumed' by that action. By using the `RewardPredictor` module and passing it this shape, the agent obtains a prediction of the reward associated with that shape. Over time, by calculating the loss between that prediction and the actual reward obtained after taking an action, the module learns to accurately predict the reward associated with every shape.[1]

### 3.3. Action Reasoner

Once the agent can predict the expected reward of its own actions, we can then guide its policy learning so that it acts in the way (it expects) will give the highest immediate reward. To achieve this, we specify an axiom to ensure that the $q-$value outputs of the Q-Network are in alignment with the predicted rewards. To achieve this, the expected reward of all the possible actions is calculated by using the `RewardPredictor` and the $q-$values are calculated by calling the `SymDQN`. Our axiom expresses the following condition: if the reward prediction of action $a_1$ is higher than the reward prediction of action $a_2$, then the $q-$value of $a_1$ must also be higher than the $q-$value of $a_2$. The learning is then guided by the LTN framework with the following formula of Real Logic used in the loss function.

$$\forall \, \text{Diag}((r_1, q_1), (r_2, q_2))(r_1 > r_2) : (q_1 > q_2) \tag{A4}$$

Two standard operators of Real Logic (Badreddine et al., 2022) are applied in this axiom: $Diag$ and guarded quantification with the condition $(r_1 > r_2)$. Firstly, the $Diag$ operator restricts the range of the quantifier, which will then not run through all the pairs from $\{r_1, r_2, q_1, q_2\}$, but only the pairs of (reward, $q$-value) that correspond to the same actions. Specifically, when $r_1$ corresponds to the predicted reward of action 'up', then $q_1$ corresponds to $q$-value of action 'up'. Secondly, we use guarded quantification, restricting the range of the quantifier to only those cases in which $(r_1 > r_2)$. If we had used implication, with the antecedent $(r_1 > r_2)$ false, the whole condition would evaluate to true. This is problematic when the majority of pairs do not fulfill the antecedent. In such a case the universal quantifier evaluates to true for most of the instances, even if the important ones, with antecedent true, are false. Guarded quantification gives a satisfaction degree that is much closer to the value we are interested in.

### 3.4. ActionFilter

Our learner can now predict the reward for each shape. For each action in a given state, it then knows what shapes could be consumed and what is their corresponding immediate reward. `ActionFilter` eliminates the actions for which the difference between their reward and the maximum immediately obtainable reward in that state is under a predefined threshold (we set it at 0.5). This allows a balance between the strictness of symbolic selection of

---

1. Fig. 4 depicts the embedding of the above-described two components in the `DuelDQN` architecture.

immediately best actions and the information about rewards available in the network as a whole. This is represented in Fig. 2. `ActionFilter` severely restricts action choice. We pre-

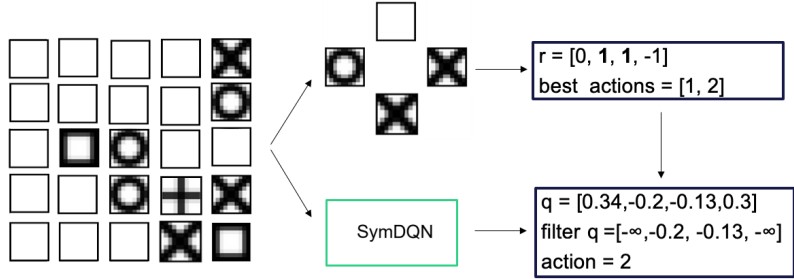

Figure 2: The process of action filtering

vented it from forcing the outcomes by switching it off during the training period. When the agent is actually running in the environment, the `ActionFilter` is used to optimize decision making. Further strategies on how this dynamic might be implemented in training must be studied, as we want to maintain the asymptotic optimality of `DuelDQNs`, while enhancing them with reasoning, when relevant.

## 4. The experiment

By comparing the baseline `DuelDQN` model with our `SymDQN` model, this study attempts to answer the following questions:

Q1 Does the `SymDQN` converge to a stable action policy faster than the baseline `DuelDQN`?

Q2 Does the `SymDQN` outperform the baseline `DuelDQN` in average reward accumulation?

Q3 Is `SymDQN` more precise in its performance than `DuelDQN`, i.e., is it better at avoiding shapes with negative rewards?

In the experiment, we analyze the impact that each individual modification has on the performance of the `SymDQN`, comparing them between each other and the baseline `DuelDQN`. We hence consider five experimental conditions:

`DuelDQN`: the baseline model-no symbolic components;

`SymDQN`: DuelDQN with `ShapeRecognizer` and `RewardPredictor`;

`SymDQN(AR)`: SymDQN with `ActionReasoner`;

`SymDQN(AF)`: SymDQN with `ActionFilter`;

`SymDQN(AR,AF)`: SymDQN enriched with `ActionReasoner` and `ActionFilter`.

Our experiment runs through 250 epochs, after which the empirically observed rate of learning of all the variations is no longer significant. Each epoch contains 50 episodes of training, and then the agent's performance is evaluated as the average score of 50 new episodes. The

score is defined as the ratio of the actual score and the maximum score obtainable in a given episode. The other performance measure we look at is the percentage of the negative-reward objects consumed.[2]

## 5. Results

In this section, we will report on the results of our ablation study. We first compare our best-performing condition, `SymDQN(AF)` with the baseline learning of the pure non-symbolic `DuelDQN`, see Fig. 5 in the Appendix. Clearly, the performance of the `SymDQN` agent equipped with `ActionFilter` is superior to `DuelDQN` both in terms of quicker convergence (high initial learning rate) and overall end performance. Secondly, we can look at different versions of our `SymDQN` to better understand what contributes to its performance. In Fig. 6 (in the Appendix) we show the performance of all four versions of `SymDQN`.

Let us now move to another performance measure: the precision of the agent in avoiding objects of the shape associated with negative rewards. We compared all five experimental conditions, see Fig. 7 in the Appendix. While the presence of `ActionReasoner` (in purple) allows a significant improvement of precision, it's the `ActionFilter` that eradicates negative rewards completely (red and green graphs). The baseline `DuelDQN` and the pure `SymDQN` perform similarly, not being able to learn to avoid negative rewards completely.

**Interpretation and Discussion**  The integration of symbolic knowledge into reinforcement learning, as demonstrated by `SymDQN`, provides several insights into the potential of neuro-symbolic approaches in AI. The ability of `SymDQN` to extract and utilize key environmental features drives a significant boost in initial learning rate and overall performance, suggesting that symbolic representations can provide a valuable advantage to neural networks, enabling them to rapidly leverage the features for better decision-making. The `ActionFilter` provides a dramatic enhancement in early-stage performance, allowing the model to make good decisions as soon as the symbolic representation is available. By leveraging the symbolic representation and understanding of the environment, `ActionFilter` prunes sub-optimal actions, aligning the agent's behavior with a symbolic understanding of the environment. The role of `ActionReasoner` is less clear: while providing a slight boost in initial performance, it hampers the overall learning rate. It seems that by forcing the model output to comply with the logical axiom, it diminishes its ability to capture information that is not described by the logical formulas.

The two components, `ActionReasoner` and `ActionFilter` use symbolic information to adjust (the impact of) $q$-values and could be seen as a form of reward shaping. On top of that, `ActionReasoner` uses (A4) to align $q$-values with predicted immediate rewards. As we can see in Fig. 6, this 'reward shaping' process is detrimental to the overall performance. A possible reason for that can be illustrated in the following example. Let's assume the agent is separated from a multitude of positive shapes by a thin wall of negative shapes. A long-term perspective of sacrificing some reward by crossing over the wall can be blocked by attaching too much value to the immediate punishment. Note, however, that although ineffective, this 'reward shaping' makes the agent more cautious/precise (see Fig. 6). While `ActionFilter`

---

2. The hardware and software specification and the hyperparameters of the experiment can be found in the Appendix A.

does not shape the reward function directly (as it is turned off in the training phase), it performs reasoning based on rewards (Fig. 2). In a given state it eliminates the possibility of executing actions for which $q$-values and immediate rewards differ too drastically.

The advantages of `SymDQN` come with trade-offs. The computational cost introduced by the additional components is non-trivial, and the logical constraints imposed on the learning might hamper performance in more complex environments. In that, the use of LTNs in reinforcement learning sheds light on the 'thinking fast and slow' effects in learning. Firstly, the use of (A1)-(A3) in `ShapeRecognizer` gives a sharp increase in the initial performance due to the understanding of the environment structure (Fig. 8). Apart from that, adjusting the reward function with `ActionReasoner` and `ActionFilter` will increase precision (as normally assumed about the System 2 type of behavior), but it can also hamper the overall performance, like it does in the case of `ActionReasoner` (see Fig. 6).

## 6. Conclusions and Future Work

This research introduces a novel modular approach to integrating symbolic knowledge with deep reinforcement learning through the implementation of `SymDQN`. We contribute to the field of Neuro-Symbolic AI in several ways. Firstly, we demonstrate a successful integration of LTNs into reinforcement learning, a promising and under-explored research direction. This integration touches on key challenges: interpretability, alignment, and knowledge injection. Secondly, `SymDQN` augments reinforcement learning through symbolic environment representation, modeling of object relationships and dynamics, and guiding action based on symbolic knowledge, effectively improving both initial learning rates and the end performance of the reinforcement learning agents. These contributions advance the field of neuro-symbolic AI, bridging the gap between symbolic reasoning and deep learning systems. Our findings demonstrate the potential of integrative approaches in creating more aligned, controllable and interpretable models for safe and reliable AI systems.

We see several potential avenues for future research. Firstly, `ShapeRecognizer` could be adapted to any grid-like environment. It could also be further developed to represent more complex environments symbolically, e.g., by integrating a more advanced object-detection component (e.g., Faster-LTN (Manigrasso et al., 2021)). With the addition of precise bounding box detection and multi-class labeling, the component could be extended to also perform hierarchical representations, e.g., recognizing independent objects and their parts or constructing abstract taxonomies. The investigation of automatic axiom discovery through iterative learning, or meta-learning, is an interesting direction that opens the doors to knowledge extraction from a model (see, e.g., (Hasanbeig et al., 2021; Umili et al., 2021; Meli et al., 2024)). Theoretically, given enough time and randomization, Q-learning converges to optimal decision policy in any environment, and so the iterative development and assessment of axioms might allow us to extract knowledge from deep learning systems that outperform human experts. While a version of `SymDQN` was shown to be advantageous, it was only tested in a single, simple environment. A broader suite of empirical experiments in more complex environments, such as Atari games or Pac-Man, is necessary to understand the generalization capabilities of the findings. These environments provide more complex and diverse challenges, potentially offering deeper insights into the advantages of `SymDQN`.

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

## Appendix A. Figures

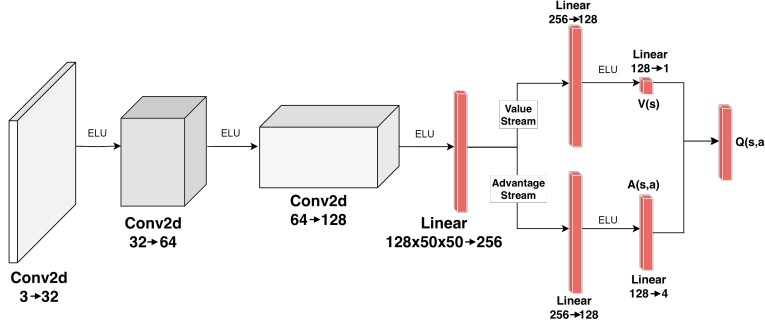

Figure 3: Network Architecture of `DuelDQN`, with the convolutional layers in white, and the dense layers in red

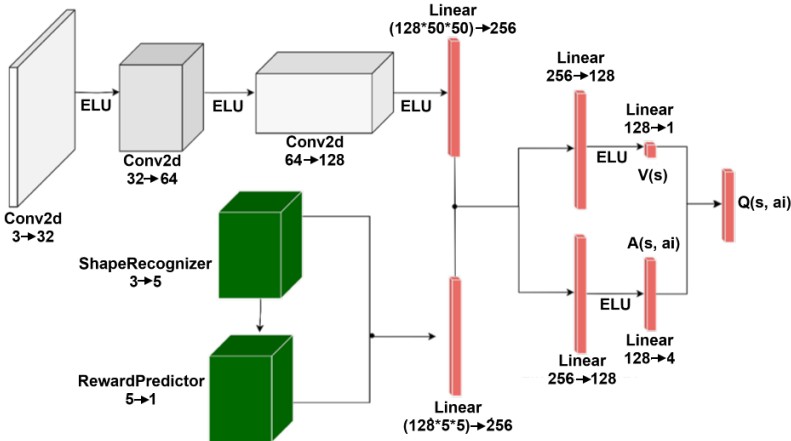

Figure 4: `SymDQN` network architecture integrating `DuelDQN` with `ShapeRecognizer` and `RewardPredictor` modules

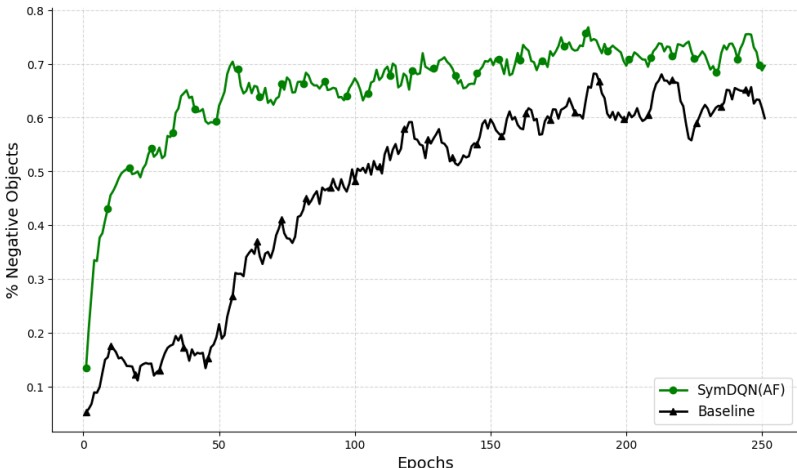

Figure 5: `SymDQN(AF)` (in green) vs. `DuelDQN` (in black): $x$-axis represents epochs and $y$-axis represents the ratio of obtained score in the episode and the maximum obtainable score in that episode.

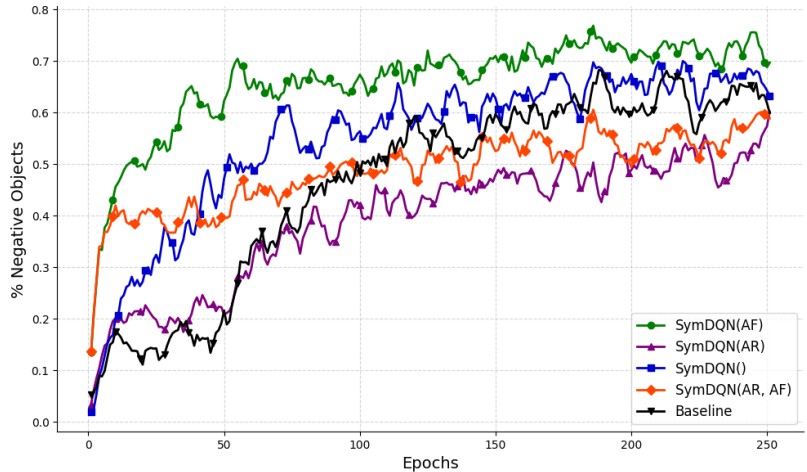

Figure 6: All versions of `SymDQN`: `SymDQN(AF)` (in green), `SymDQN(AR,AF)` (in red), `SymDQN` (in blue) and `SymDQN(AR)` (in purple); the $x$-axis represents epochs and the $y$-axis represents the ratio of obtained score in the episode and the maximum obtainable score in that episode. We report in standard deviations in the Appendix.

## Appendix B. Specifications

**Hardware and software specifications**   The experiments, hyper-parameter tuning, and model variation comparisons were performed on a computing center machine (Tesla V100 with either 16 or 32GB). The coding environment used was Python 3.9.5 and Pytorch 2.3.1 with Cuda Toolkit 12.1. To integrate LTNs, the LTNtorch (Carraro, 2022) library was used,

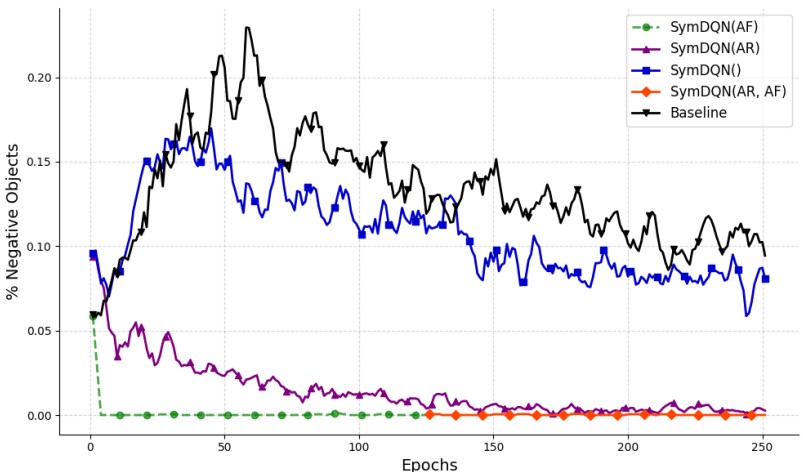

Figure 7: Agent's precision in all conditions: `SymDQN(AF)` (in green), `SymDQN(AR,AF)` (in red), `SymDQN(AR)` (in purple); `SymDQN` (in blue); `DuelDQN` (in black): the $x$-axis represents epochs and the $y$-axis represents the percentage of negative-reward objects consumed by the agent. Note that the green and red lines overlap.

a PyTorch implementation by Tommaso Carraro. For experiment tracking, the ClearMl Platform (cle, 2023) was used. The code is provided as a separate .zip file.

**Hyperparameters** The hyper-parameters used for training were: explore steps: 25000, update steps: 1000, initial epsilon: 0.95, end epsilon: 0.05, memory capacity: 1000, batch size: 16, gamma: 0.99, learning rate: 0.0001, maximum gradient norm: 1. Semantics of quantifiers in LTN used $p = 8$.

## Appendix C. Standard deviations

Performance was assessed through 5 independent experimental runs per configuration. Each run evaluated 50 distinct environments to calculate average scores per epoch; ($*$) stands for SymDQN($*$), values for epochs 50, 150, 250 (smoothed, rolling window 5). First table shows overall performance (rewards, r) and standard deviations (sd). Second table shows the ratio of negative shapes collected (inverse of precision, p) and sd.

| Model | 50r | 50sd | 150r | 150sd | 250r | 250sd |
|-------|------|------|------|-------|------|-------|
| (AF) | 0.65 | 0.03 | 0.70 | 0.01 | 0.71 | 0.01 |
| (AR) | 0.26 | 0.03 | 0.47 | 0.01 | 0.53 | 0.04 |
| () | 0.43 | 0.05 | 0.65 | 0.02 | 0.66 | 0.02 |
| (AR,AF) | 0.40 | 0.02 | 0.53 | 0.02 | 0.59 | 0.02 |
| DDQN | 0.24 | 0.06 | 0.57 | 0.02 | 0.64 | 0.01 |

Table 1: Overall performance (rewards, r) and standard deviations (sd)

| Model | 50p | 50sd | 150p | 150sd | 250p | 250sd |
|---|---|---|---|---|---|---|
| (AF) | 0.00 | 0.00 | 0.00 | 0.00 | 0.00 | 0.00 |
| (AR) | 0.02 | 0.00 | 0.01 | 0.00 | 0.00 | 0.00 |
| () | 0.17 | 0.02 | 0.12 | 0.02 | 0.08 | 0.00 |
| (AR,AF) | 0.00 | 0.00 | 0.00 | 0.00 | 0.00 | 0.00 |
| DDQN | 0.19 | 0.03 | 0.14 | 0.01 | 0.10 | 0.00 |

Table 2: The ratio of negative shapes collected (inverse of precision, p) and sd

