# OpenReview forum: "SymDQN: Symbolic Knowledge and Reasoning in Neural Network-based Reinforcement Learning"
_nesyconf.org/NeSy/2025/Conference_Phase_2 — NeSy 2025 - Phase 2 Poster_

### Official Review · Reviewer_PBQp · 2025-07-07
**Interesting idea with limited experiment**

**Rating:** 7
**Confidence:** 4

**Review:**

The paper proposes SymDQN, a modular framework that integrates Logic Tensor Networks (LTNs) into the Dueling Deep Q-Network (DuelDQN) architecture. The aim is to incorporate symbolic reasoning into deep reinforcement learning to enhance interpretability, control, and learning efficiency. Key symbolic modules include shape recognition, reward prediction, action reasoning, and action filtering. The framework is evaluated on a custom grid-world environment, demonstrating faster convergence and better precision compared to the baseline.

Strengths
	•	A novel approach to integrate neurosymbolic into Reinforcement Learning, specifically with dual deep Q-Net.
	•	The experiment show promising results.
	•	Methodology and logic formalism (Real Logic) are clearly explained and well-motivated.

Weaknesses
	•	Limited environment: Evaluation is restricted to a single, simple 5×5 grid; unclear how the framework generalises to more complex or continuous environments. est on more complex RL environments (e.g., Atari, MiniGrid) to assess scalability and generalisation.
	•	Computational overhead: Symbolic components add complexity; potential scalability issues not addressed. Provide runtime or resource usage comparisons to highlight the computational trade-offs.

 The paper offers a meaningful contribution to neuro-symbolic RL, with solid methodology and clear results. Wider validation and computational analysis would strengthen its impact.

**Anonymity:**

Remain anonymous

---

### Official Review · Reviewer_oRSG · 2025-07-08
**Review of SymDQN: Symbolic Knowledge and Reasoning in Neural Network-based Reinforcement Learning**

**Rating:** 8
**Confidence:** 4

**Review:**

Summary:
The authors extend/apply LTNs to a reinforcement learning problems.
The integration of LTNs in the training process allows agents to use logic to learn the structure of the environment, to predict how different objects in the environment interact with each other, and to
guide its actions by performing elementary reasoning about rewards.

Paper structure assessment:
Section - Introduction and background:
solid, not missing contextualization.

Section - Related work:
focus on LTN and real logic, "the basic functioning of LTN". Uses 3 examples to make the point. Possibly did not need all examples and material could be a subsection.

Section - Methodology:
proposed experiments with custom Gymnasium (Towers et al., 2024 - cited in the paper) and the environment ShapesGridEnv was designed for the experiments in (Badreddine and Spranger,2019 - cited in the paper).
Choice of environment justified by simplicity and to allow comparison with  (Badreddine and Spranger, 2019).
The authors start experiments with the DuelDQN architecture; extended with new symbolic components: shape recognition (ShapeRecognizer), reward prediction (RewardPredictor), action reasoning (ActionReasoner), and action filtering (ActionFilter).
Sections 3.1-3.4 explain these new symbolic components and their use in the experiments of section 4.
Particularly relevant are A1-A3 which represent the first instance of actual knowledge injection in the system.

Section - Experiments (succint section describing the questions tackled in the paper):
The authors try to answer three questions:
Q1 Does the SymDQN converge to a stable action policy faster than the baseline DuelDQN?
Q2 Does the SymDQN outperform the baseline DuelDQN in average reward accumulation?
Q3 Is SymDQN more precise in its performance than DuelDQN, i.e., is it better at avoiding shapes with negative rewards?
The authors consider 5 experimental conditions:

DuelDQN: the baseline model-no symbolic components;

SymDQN: DuelDQN with ShapeRecognizer and RewardPredictor;

SymDQN(AR): SymDQN with ActionReasoner;

SymDQN(AF): SymDQN with ActionFilter;

SymDQN(AR,AF): SymDQN enriched with ActionReasoner and ActionFilter.

Section - Results:
Authors state that the integration of symbolic knowledge into reinforce-
ment learning, as demonstrated by their SymDQN framework provides several insights into the potential of neuro-symbolic approaches.
SymDQN extracts and utilizes key environmental features and drives a significant boost in initial learning rate and overall performance.
The authors discuss the trade-offs of their approach:

1)  The computational cost introduced by the additional components is non-trivial;
2)  the logical constraints imposed on the learning might hamper performance in more complex environments.
3)  The use of LTNs in reinforcement learning sheds light on the ‘thinking fast and slow’ effects in learning, though this point deserves more thoughtful analyses, not only based on A1-A3 effects.

Overall, this is a solid paper that introduces an interesting approach to validate neurosymbolic approaches based on LTN extended with RL components.

**Anonymity:**

Remain anonymous

---

### Official Review · Reviewer_REF9 · 2025-07-08
**A case study that combines symbolic expressions with reinforcement learning.**

**Rating:** 4
**Confidence:** 4

**Review:**

The authors study a mix of symbolic representations and reinforcement learning, so as to improve the latter with insights coded by the former. This is certainly relevant to a discussion of neuro-symbolic techniques. The paper is overall easy to follow and there are no problems with the quality of the text. However, the significant of the results is somewhat narrow; while some statements in the paper seem to advertise a novel framework to combine symbolic and neural methods through a particular logic (Real Logic), the paper actually describes a case study that employs mostly well known techniques to a reasonably controlled setting.

One problem here is that, despite the fact that authors describe Real Logic, and give short examples, the reader is left without a clear sense as to why this particular logic is appropriate here. Presumably the selection of this logic is a major point. Or is it not? If not, then why not use a more common language, even first-order logic or perhaps some probabilistic logic? What can be expressed with this language that cannot be expressed with others? It would be nice to be clear about these points.

The first sentence of Section 6 is an example of a statement that seems too strong. The paper seems to describe a case study, but the sentence there states that a novel modular approach has been presented. The approach is then to use logical expressions in Real Logic to help the neural subsystems? Isn't this something that Real Logic was supposed to offer, a combination of logical expressions and numeric manipulation? Please clarify.

Concerning the text, here are a couple of suggestions:
- The paragraph "Related Work" in page 2 contains too many distinct points, all presented without any pause. It is something to improve.
- Mathematical expressions should end with period/comma as appropriate.

**Anonymity:**

Remain anonymous